# Translation, Cross-Cultural Adaptation, and Psychometric Properties of the Polish Version of the Hip Disability and Osteoarthritis Outcome Score (HOOS)

**DOI:** 10.3390/medicina55100614

**Published:** 2019-09-20

**Authors:** Wojciech Glinkowski, Agnieszka Żukowska, Małgorzata Dymitrowicz, Emilia Wołyniec, Bożena Glinkowska, Dorota Kozioł-Kaczorek

**Affiliations:** 1Centre of Excellence “TeleOrto” for Telediagnostics and Treatment of Disorders and Injuries of the Locomotor System, Department of Medical Informatics and Telemedicine, Medical University of Warsaw, 00-581 Warsaw, Poland; a.zukowska@yahoo.co.uk; 2Polish Telemedicine and eHealth Society, 03-728 Warsaw, Poland; malgorzata.reh.lecz@interia.pl (M.D.); emi.wolyniec@gmail.com (E.W.); boglinkowska@o2.pl (B.G.); dorota.k.kaczorek@gmail.com (D.K.-K.); 3Rehabilitation Department, Central Military Medical Clinic, 00-671 Warsaw, Poland; 4Department of Rehabilitation, Second Faculty of Medicine with the English Division and the Physiotherapy Division, Medical University of Warsaw, 02-109 Warszawa, Poland; 5Department of Sports and Physical Education, Medical University of Warsaw, 02-109 Warsaw, Poland; 6Faculty of Economic Sciences, Warsaw University of Life Sciences, 02-787 Warsaw, Poland

**Keywords:** hip osteoarthritis, disability, quality of life, translation, questionnaire-based research

## Abstract

*Background and Objectives:* This study aimed to translate the Hip disability and Osteoarthritis Outcome Score (HOOS) into the Polish language, to determine its validity and reliability, and to assess its main psychometric properties. *Materials and Methods:* A total of 332 hip osteoarthritis (OA) subjects were recruited to the study group and 90 healthy subjects to the control group. The study consisted of the HOOS translation and the assessment of the discriminative power, internal consistency, and the potential floor and ceiling effects followed by the determination of the construct validity and test-retest reliability. The analysis was performed using Western Ontario and McMaster Universities osteoarthritis index (WOMAC) and SF-36 questionnaires. *Results:* The translation process consisted of forward translation, reconciliation, backward translation, review, harmonization, and proofreading. The hip OA patients reported a reduced HOOS score when compared to the control subjects. The discriminant validity of the questionnaire was confirmed. A Cronbach’s alpha of 0.97 was found, indicating a high internal consistency. The HOOS showed a significant correlation with the SF-36 and WOMAC, which ranged from *r* = −0.93, *p*-value < 0.05 for WOMAC total score to *r* = 0.92, and *p*-value < 0.05 for WOMAC daily living. No floor or ceiling effects were found. A very high intraclass correlation coefficient (ICC) was found (0.93–0.97) for the total score and the individual domains of the HOOS. *Conclusions:* The Polish HOOS is valid and reliable for evaluating the outcomes of hip OA patients in Poland. This questionnaire may be used with confidence for clinical and research purposes.

## 1. Introduction

Osteoarthritis (OA) degenerates the cartilage and surrounding tissues, which leads to joint pain, stiffness, limited range of motion, and a deterioration of the quality of life. These are the main clinical symptoms of OA. OA is the most common joint disorder worldwide and primarily affects the knees, hips, and spine [1,2]. OA is highly prevalent globally [2,3,4]. Hip OA is regarded as a progressive decrease of the hip joint function and pain and its function with aging [2,4,5,6] has a significant impact on the health-related quality of life (HRQoL) [7,8,9,10,11,12]. Depending on the severity of OA, conservative treatment, or surgical intervention, can be performed [13]. The end-stage of OA requires total hip arthroplasty (THA) to alleviate severe pain and disability [14,15]. 

Several Patient Reported Outcome Measures (PROMs) were developed for the specific assessment of the OA patients, e.g., the Western Ontario and McMaster Universities osteoarthritis index (WOMAC), Musculoskeletal Health Questionnaire (MSK-HQ), and Harris Hip Score (HHS) [16,17,18,19].The instruments used to assess HRQoL are based on patients’ opinions (patient-reported outcomes measures (PROMs)) and focus on the assessment of physical, mental, and social functions as well as well-being. The Hip disability and Osteoarthritis Outcome Score (HOOS) [20,21] is patient-friendly and provides an option to examine self-reported changes of status in pain, activities of daily living, sport and recreation, and hip-related quality of life (QoL). HOOS is considered appropriate for more active and younger patients than the WOMAC, due to added subscales. HOOS may be used over short-term and long-term intervals to assess changes induced by treatment. The HOOS was developed in 2003 by the Swedish researchers to extend the WOMAC [20,21,22] and to evaluate the patients’ view about their hip dysfunctions. The HOOS questionnaire has been developed, validated, and translated into several languages (Chinese, Dutch, French, German, Italian, Japanese, Korean, Persian, Spanish, Thai), which makes the opportunity to conduct international, multicenter studies [20,21,23,24,25,26,27,28]. Questionnaires are available to download free of charge from www.koos.nu. The scaling system of the HOOS has been validated, and its responsiveness has been confirmed in patients with hip osteoarthritis in relation to medical and surgical treatments [24,29].

The HOOS can discriminate hip OA subjects from non-hip ones with regard of their health-related QoL [30]. It is necessary to validate and cross-culturally adapt the questionnaire to ensure its cultural equivalence and suitability in different populations. In general, terms translating from the original version into other languages may improve establishing an essential instrument for clinical purposes. Translated versions of the questionnaire should be approved and standardized. To our knowledge, there was a gap in this methodological approach for a specific hip questionnaire in Polish that measures not only symptoms and function but also hip-related QoL. This study aimed to translate the English version of the HOOS questionnaire into a conceptually similar Polish language, to determine its validity and reliability, and to assess its main psychometric properties.

## 2. Materials and Methods

### 2.1. Participants and the Protocol

A total of 332 subjects were recruited into the study group (patients qualified for hip surgery) and 90 subjects were recruited into the control group (healthy patients) in outpatient clinics in Poland (Warsaw), which comprised a large group of patients suffering Hip OA and healthy volunteers. The admission of younger volunteers to the control group was decided due to the need to compare with a young group of adults with reference to good hip function. The age-matched group was not employed due to the expected non-specific impairment of the musculoskeletal system. All participants were given informed consent to participate in the study. Patients and volunteers were informed about the objectives and rules of participation and procedures in their course. Before recruitment, patients received concise information on the main problems associated with the course of osteoarthritis of the hip.

Ethical issues were considered. Institutional review board approval was obtained for the study (approval No. KB/157/2009, 25 August 2009), and investigations were carried out following the rules of the Declaration of Helsinki of 1975. General demographic variables analyzed included: age, gender, and anthropometric parameters (body mass, height). Recruited patients/subjects were interviewed to confirm their abilities for participation in the study.

The inclusion criteria for the hip OA group, according to previous studies [31], were:capability to participate following outpatient treatment (three days per week minimum),motivated to participate in the study,fluent in the Polish language,35 years of age and older,primary impairment diagnosis of osteoarthrosis of the hip,clinically and radiologically confirmed the diagnosis of the hip OA.

Participants were excluded from the study with the inability to: to cooperate, understand and fulfill the questionnaires, understand the Polish language, have other inabilities to participate in the study (i.e., medical conditions, being alcohol or substance dependent within six months, or current alcohol or substance abuse, with an amputated limb, cardiac or other medical instability, immobilized, fractured, having active malignancy, and mental illness).

### 2.2. Procedures

#### 2.2.1. Polish Translation

The study aimed to develop the Polish translation of the HOOS and to evaluate its psychometric properties (internal consistency, test-retest reliability, etc.), along with its interpretability and acceptability, according to the protocol. The data were anonymized during the study to keep compliance with the legislation on personal data protection.

#### 2.2.2. Psychometric Validation of the Polish Version of the HOOS

The validation of the psychometric properties of the Polish version of the HOOS questionnaire followed the recommendations [17]. The study consisted of the assessment of the discriminative power, internal consistency, and the potential floor and ceiling effects followed by determining the construct validity and test-retest reliability. The hip OA patients were compared with non-osteoarthritic (non-OA) subjects based on the symptoms. The analysis was performed using commonly accepted questionnaires (WOMAC and SF-36—Polish version [32]) that have similar dimensions. A QoL with poorer results was anticipated in osteoarthritic patients when compared to the control. 

Participants were asked to fill in questionnaires at the outpatient clinic, namely the Polish HOOS, the SF-36, and the WOMAC. For stable patients, test-retest typically requires a time interval of 2–14 days [33]. The hip OA subjects were asked to re-fill the questionnaire, after a one-week break, to analyze the stability of the HOOS Polish version questionnaire. The results of 45 subjects who did not report any significant health changes during this one week were analyzed.

#### 2.2.3. Questionnaires

##### HOOS

The HOOS consist of 40 items divided into five domains: pain (10 items), symptoms including stiffness and range of motion (five items), activity limitations-daily living (17 items), sport and recreation function (four items), and hip-related quality of life (four items). The Likert Scale is utilized to answer all items. The patient’s responses are based on the five-point Likert scale (none, mild, moderate, severe, extreme). Each answer is scored from 0 (no problems) to 4 (extreme problems), and the final score is calculated using the formula given in the user’s manual [21,26,28,34]. 

##### SF-36

The self-administered generic health status questionnaire (SF-36) was applied to test for convergent construct validity [35]. It is a widespread, validated self-administered psychometric generic questionnaire on HRQoL, which is also available in Polish [32]. The instrument contains 36 items grouped into eight multi-item scales: physical functioning (10 items), social functioning (2 items), role limitations due to physical problems (4 items), role limitations due to emotional problems (3 items), mental health (5 items), energy/vitality (4 items), pain (2 items), and general perception of health (5 items). The score can assess health status according to scores ranging from zero to 100. Higher scores indicate better health: 0 indicates the worst health status and 100indicates the best possible health status.

##### WOMAC

The Western Ontario and McMaster Universities Osteoarthritis Index (WOMAC) is a patient–reported outcome measure for populations with hip and knee OA or Total Joint Arthroplasty [26]. The WOMAC is a multidimensional scale of 24 items grouped into three subscales: physical function (17 items), pain (5 items), and stiffness (2 items). The Likert scale allows five response levels for each item (scored 0–4) representing different degrees of intensity (none, mild, moderate, severe, or extreme). The data for each subscale are standardized to a range of 0–100, where 0 is the best and 100 is the worst health status [22]. 

### 2.3. Statistical Analysis

All the analyses were carried out using TIBCO Software Inc., Palo Alto, CA 94304, USA (2017) Statistica (data analysis software system), version 13. Such variables including gender, age, height, and weight were treated as descriptive variables, i.e., variables which identify the examined group of patients. Answers to items were defined as study variables. The results were considered statistically significant at the significance level α = 0.05 if the critical level *p*-value < 0.05. Normality of quantitative variables was tested by the Shapiro–Wilk test.

Quantitative variables with a normal distribution were expressed by the mean and standard deviation (mean ± SD). Quantitative variables, which showed a non-normal distribution, were expressed as a median (Me), a first quartile, and a third quantile, i.e., Me (P25−P75). Qualitative variables were reported as absolute and relative frequencies (%).

Differences in characteristics between the hip OA group and control group were tested with the parametric Student’s *t*-test or the non-parametric Mann–Whitney U test for quantitative variables. The test was chosen due to the distribution of variables. Quantitative variables with the normal distribution were tested with the *t*-test, and the quantitative variables, which did not show normal distributions, were tested with the Mann-Whitney U test. The differences between means of quantitative variables from normal distribution were estimated in confidence intervals on confidence level 1 − α = 0.95 (95% CI). The Cronbach alpha coefficient was applied for an internal consistency assessment. The high level of the internal consistency coefficient is indicated by a value greater than 0.70. Floor and ceiling effects were evaluated to find significant ones if higher than 15%. These effects were defined when a high percentage of the population had the lowest or the highest score, respectively. The construct validity was investigated only for hip OA participants by measuring the convergent validity. The construct validity can be defined as the degree to which the questionnaire measures what it claims or allegedly measures. Construct validity is demonstrated by using correlations to verify the relevance of the questionnaire’s elements. The reliable items from the HOOS questionnaire were assessed to determine the construct validity. Highly correlated scores are called convergent validity. The presence of the convergent validity means that the construct validity is supported. The correlation of each domain with the total score of the HOOS was assessed using Spearman’s correlation coefficient (all data were not normally distributed). The correlation between the HOOS and the other questionnaires was also assessed using Spearman’s correlation coefficient (all data were not normally distributed). The intraclass correlation coefficient (ICC) was used to assess reliability. Values ICC are confined to the interval [−1,1], where ICC ≈ 1 means very high reliability.

## 3. Results

### 3.1. Translation Process 

The translation process consisted of preparation, forward translation, reconciliation, backward translation, review and harmonization, and proofreading, according to the recommendations [36,37]. The COnsensus-based Standards for the selection of health status Measurement INstruments (COSMIN) guidelines and checklists were used to verify the complete translation and validation process [37,38]. 

#### 3.1.1. Forward Translation

Two independent translations from English to Polish were obtained from two translators who are Polish native speakers, a physiotherapist (AŻ), and an orthopedic surgeon (WG), independently translated the questionnaire. Then, a consensus meeting was organized so that the two translators could meet and agree upon a single shared version. The synthesis process of the initial translations provides a single and unified translation.

#### 3.1.2. Back-Translation

Two independent backward translations into English were performed blindly to the HOOS original version. Two independent English back translations were done blindly to the original HOOS version by native English translators, who are fluent in the target language. The translator did not see the original English source items or item definitions. The task of reverse translation was to obtain a translation that must reflect what the translation into the target language says without embellishing it.

#### 3.1.3. Review and Harmonization

A multidisciplinary committee included two translators (AŻ and WG) and another physiotherapist (BG) with a documented expertise in questionnaire validation reviewed and made the back-translation consistently. No significant difficulties during the translation process were faced. The consensus was achieved for some minor discrepancies, which reflects the cultural context and the spoken Polish language specificity. 

Five hip OA patients have tested the pre-final version to ensure understanding the purpose and meaning of each question to provide the final Polish version of the HOOS. 

The translation of the current Polish version of SF-36 [32,39] served as an example of translation and to discuss the final version. The final agreement was accomplished on items of a questionnaire that were checked against understandability and transcultural adaptation.

#### 3.1.4. Proofreading 

As mentioned above, committee reviewed translation history (forward translations, reconciled versions, backward translations, and pre-final version) with the original questionnaire resulting in a final of the Polish translation of the questionnaire. The team collected a whole set of discussed issues concerning translation (terms, wording, cultural appropriateness) in order to use it for future similar work. All items of the original HOOS questionnaire were translated flawlessly, without major substantial problems. The latest version was administered to a subgroup of five hip OA patients for cognitive debriefing. Patients expressed their opinions on used wording, understandability, interpretation, and cultural relevance of the translation. The ready-to-use version of Polish HOOS was released after the final review.

### 3.2. Descriptive Analysis

A total of 332 subjects were enrolled in the study group (patients qualified and waiting for hip surgery) and 90 subjects to the control group (healthy patients). Patients were asked to complete the HOOS questionnaire (see Appendix A), the WOMAC questionnaire, and the SF-36 questionnaire. There were 188 (57%) women and 144 (43%) men in the hip OA group, while there were 56 (62%) women and 34 (38%) men in the control group. Hip OA patients were waiting for a total hip replacement—78 (24%) patients with the right hip (51 women, 27 men), 111 patients with the left hip (67 women, 47 men), and 143 patients with both hips (73 women, 70 men). The mean age of the subject in the study group was 61.70 ± 10.57 years, with the mean age of woman being 63.14 ± 11.00 years, and the mean age of men was 59.83 ± 9.70 years. The mean age of the subject in the control group was 37.05 ± 17.10 years, with the mean age of woman being 38.20 ± 17.77 years, and the mean age of men was 35.18 ± 16.02 years. Under the research premises, the hip OA patients were significantly older than subjects of the control group (*p*-value = 0.00). The 95% confidence interval for the difference of mean age was CI = (−27.51, −21.79) years. The hip OA group demonstrated a significantly higher Body Mass Index (BMI) than the control group (*p*-value = 0.00). The BMI of the hip OA was 27.54 (24.68–30.12), and the BMI of the control group was 23.36 (20.70–26.23). The analysis of the normality of the study variables showed that none of these variables have a normal distribution. Thus, in further analysis, it was necessary to use non-parametric techniques.

### 3.3. Psychometric Validation

#### 3.3.1. Discriminative Power

The hip OA patients reported a reduced HOOS global quality of life and other domains compared to the control subjects (Table 1). The discriminant validity of the questionnaire was confirmed.

#### 3.3.2. Internal Consistency

A Cronbach’s alpha of 0.94 was found, which indicates a high internal consistency. Deleting the domains one at a time led to Cronbach’s alpha values varying between 0.91 (when deleting HOOS daily living) and 0.94 (when deleting HOOS quality of life). Moreover, all domains showed a significant correlation with the total score of the HOOS, which ranges from *r* = 0.84, *p*-value < 0.05 (for HOOS quality of life) to 0.95, and *p*-value < 0.05 (for HOOS daily living) (see Table 2).

#### 3.3.3. Construct Validity

The construct validity was investigated by measuring the convergent validity. Correlation between the HOOS questionnaire and the WOMAC and SF-36 questionnaires, which were supposed to have similar dimensions, were assessed. As expected, the HOOS questionnaire showed significant correlation with the SF-36 and WOMAC, ranging from *r* = −0.93, *p*-value < 0.05 for WOMAC total score to *r* = 0.92, and *p*-value < 0.05 for WOMAC daily living. Results of the construct validity assessment are presented in Table 3.

#### 3.3.4. Floor and Ceiling Effects

No hip OA patients presented with the lowest score to the questionnaire (0 points) or the maximal score (100 points). Therefore, neither floor nor ceiling effects were found for the questionnaire.

#### 3.3.5. Test-Retest Reliability

The sample of 45 subjects agreed to complete the HOOS questionnaire after seven days. The global score of the HOOS moved a little from 221.47 (133.01–287.57) to 207.72 (136.69–2 70.88), which reflected the moderate stability of the questionnaire across time. Excellent agreement occurred between the test and the retest. Results are presented in Table 4. For both the total score and for the individual domains of the HOOS, a very high ICC was found (from 0.93 for HOOS quality of life to 0.97 for HOOS daily living—reflecting perfect reliability).

## 4. Discussion

In recent years, the need for translation and adaptation of intercultural instruments of patient-reported outcomes rapidly increased, which were developed in English-speaking countries for use in international clinical trials [40,41,42,43,44]. This study aimed to translate the original English HOOS questionnaire following international guidelines into a conceptually equivalent Polish version [26,40,44,45,46]. Improved general health and increased life span rise higher expectations on physical activity and function by the patients undergoing treatment [9,47]. The study also aimed to show the presence of the floor and ceiling effect for the control group and the lack of floor and ceiling effect for a hip OA osteoarthritis group. We assumed the importance of measuring patient’s expectations using valid and reliable measures for assessing the QoL of patients with hip OA [18]. Cross-cultural adaptations of measures are needed to enable standardization between cultures and countries [24,25,26,28,41,42,48,49].

Since the original version of HOOS was first developed and approved in 2003 [21], many language versions of the questionnaire became available, which allowed researchers to conduct international, multicenter studies among patients suffering from hip OA [23,24,25,26,27,28,41,42,45,50]. We collected a relatively large sample of patients with advanced hip OA, due to a specific interest of the orthopedic department treating hip OA patients. These patients may have been more informed and knowledgeable regarding the disease, which was already at the invitation stage. Due to close cooperation with patients, all questionnaires were filled out, and the lack of missing data may reflect good acceptance and feasibility of the Polish HOOS. Missing data is rarely described in the studies [25,26,49].

The English version of the HOOS questionnaire was obtained [34]. The first attempt to develop the first Polish translation of HOOS began in 2008 shortly after meeting inaugurating the “Clinical Leading Environment for the Assessment of Rehabilitation protocols in home care” (CLEAR) project funded under the ICT Policy Support Program—CIP-ICT-PSP-2007.2.2—ICT for Aging Well [31,51]. The HOOS underwent a validation process in which it demonstrated appropriate psychometric properties for the clinical and research application, comparable to other studies. The data required for psychometric testing was collected right before the beginning of the clinical part of the CLEAR study (2010).

Evidence of an equivalence between Polish and English versions of the questionnaire was provided by the high consistency with the original, internal consistency of the HOOS (Cronbach’s alpha of 0.9), which appeared comparable with the original version. The respective internal consistency is comparable to other language versions of the HOOS – simplified Chinese [52], Persian [24], Italian [28], Dutch [4], German [26], Thai [25], and others. For the total score and for the subscales of the HOOS, a very high ICC was found reflecting excellent reliability. The Polish HOOS showed a strong and significant correlation with the SF-36 and WOMAC. The Polish HOOS translation demonstrated excellent psychometric properties. The Polish version appears to be useful for evaluating the patient-relevant outcome of hip OA patients. The psychometric properties of the Polish version of the questionnaire were assessed and considered satisfactory and similar to the original HOOS. Patients with OA of the hip joint showed a significantly reduced global QoL compared with controls without OA, as anticipated. The discriminant validity of the Polish version of the questionnaire was confirmed. Similar observations have been made in other HOOS translations [23,24,25,26,27,28,41,42,45]. The construct validity was investigated by measuring the convergent validity [23,28,45,53]. In our study, the hip OA patients had a significantly higher weight than non-OA ones (*p*-value < 0.05). The control group was much younger than the study group, which helped researchers understand the response ranges in the group not affected by osteoarthritis. The metric variables were significantly different for the control group to set references for young adults instead of age-matched controls that were typically related to OA as anticipated.

The control group responded reliably due to the lack of symptoms in this group. Attention should be paid to the control group as a model of a healthy population without symptoms, which is an ideal comparison for patients with osteoarthritis.

The described approach has been accepted to delineate the extent of severity in osteoarthritis of the hip and show how seriously the hip OA affects the domains reflected in the HOOS questionnaire. In this study, the lowest Cronbach’s alpha found for the QoL subscale in the Swedish version of HOOS [21], and the Pain subscale in the Dutch version [27], was not present. In line with other validation studies, the highest Cronbach’s alpha was found for the functioning daily living subscale. All HOOS domains correlated strongly with WOMAC, with higher correlations between scales that are intended to measure similar constructs, as expected. Sufficient strength of correlations was also found for SF-36 domains. A high correlation between HOOS and WOMAC questionnaires can be explained by their specific development for the outcome measurements of hip OA. The lower correlation between HOOS and SF-36 can be explained by the less specific items in SF-36 used for assessing the quality of life in various medical conditions, not only the hip OA. Since the three scores have a different focus and target population, their contents may not be easy to compare them to each other in all aspects. However, the lower correlation may not necessarily mean lower validity. The reliability estimates of this study suggest that the Polish version of HOOS is internally consistent and stable over time. All subscales met criteria for internal reliability. Test-retest reliability and internal consistency were very good for both the overall result and the partial results. The Polish version showed similar values of Cronbach alpha for all subscales except for the hip-related QoL subscale, which was higher in the present study compared to that of the original HOOS [21]. The Polish HOOS demonstrated strong convergent construct validity. Compared to the findings of Nilsdotter et al. [21], the present study’s values for the physical functioning subscale of the SF-36 versus the HOOS subscales were similar.

The absence of any floor or ceiling effects in the hip OA group of the hip indicates that the original HOOS contains items that capture patients at both the lower and upper end of the scale, and that the test sample was balanced in terms of severity of symptoms and/or impairments.

### 4.1. Study Strengths

The strengths of this study include standardized methods used in all procedures and a large and homogeneous group of patients with OA of the hip joint. The HOOS questionnaire was developed, especially for patients with osteoarthritis of the hip, and this is especially important for this group of patients. We assume an additional strength of the study is passing the rigorous translation. The patient and healthy subjects did not mention questionable items of the questionnaire in any subscale, unlike other studies [26].

### 4.2. Limitations

This study has some limitations because participants may not represent the entire spectrum of patients with OA of the hip since they were recruited from an outpatient clinic focused on the OA hip. Eligible patients for THR likely had more severe OA symptoms. The use of a homogeneous group of severe, older, and often overweight hip OA cases may also limit the discriminatory power of the instrument. The severity of symptoms in the study group can also be considered a limitation. However, differences in the severity of symptoms assessed based on questionnaire responses varied.

## 5. Conclusions

In conclusion, the Polish HOOS questionnaire scored high test-retest reliability, high internal consistency, and relevant construct validity (i.e., comparable to another language version of the HOOS). The Polish HOOS questionnaire translation may be considered valid and reliable for evaluating the outcomes of hip OA patients and osteoarthritis-related treatments in Poland. The presented Polish version of the HOOS questionnaire can be used with confidence for clinical and research purposes as an equivalent to the original English version. Further studies may consider additional psychometric testing, including responsiveness to change and calculation of the minimal important difference.

## Figures and Tables

**Table 1 medicina-55-00614-t001:** The discriminative power of the HOOS questionnaire.

HOOS Score	The Hip OA Group (*n* = 332) Me (P25–P75)	The Control Group (*n* = 90) Me (P25–P75)	*p*-value
symptoms/stiffness	50.00 (35.00–65.00)	100.00 (95.00–100.00)	<0.05
pain	47.92 (35.00–67.50)	100.00 (100.00–100.00)	<0.05
function, daily living	45.59 (27.94–63.23)	100.00 (100.00–100.00)	<0.05
function, sports and recreational activities	25.00 (6.25–43.75)	100.00 (100.00–100.00)	<0.05
quality of life	25.00 (18.75–43.75)	100.00 (100.00–100.00)	<0.05
total	194.67 (133.68–275.00)	500.00 (495.00–500.00)	<0.05

**Table 2 medicina-55-00614-t002:** Correlations of the HOOS total score with individual domains.

HOOS Score	HOOS Total Score, *r*	*p*-value
symptoms/stiffness	0.87	<0.05
pain	0.92	<0.05
function, daily living	0.95	<0.05
function, sports and recreational activities	0.89	<0.05
quality of life	0.84	<0.05

**Table 3 medicina-55-00614-t003:** Correlations of the HOOS total score with other questionnaires.

Variable	HOOS Total Score, *r*	*p*-value
WOMAC score pain	0.86	<0.05
WOMAC score_stiffness	0.76	<0.05
WOMAC score_function, daily living	0.92 *	<0.05
WOMAC score_total	−0.93	<0.05
SF-36 score_physical functioning	0.71	<0.05
SF-36 score_role-physical	0.46	<0.05
SF-36 score_role-emotional	0.43	<0.05
SF-36 score_vitality	0.49	<0.05
SF-36 score_mental health	0.44	<0.05
SF-36 score_social functioning	0.60	<0.05
SF-36 score_bodily pain	0.64	<0.05
SF-36 score_general health	0.35	<0.05
SF-36 score_health now/before a year	0.45	<0.05

* The total score outcome measure is transformed in a worst to best scale (100—no symptoms, 0—extreme symptoms).

**Table 4 medicina-55-00614-t004:** The results of the test-retest reliability of the HOOS Polish version.

HOOS Score	Test	Retest	ICC	95% CI
symptoms/stiffness	50.00 (35.00–70.00)	50.00 (30.00–65.00)	0.94	0.89–0.97
pain	50.00 (37.50–82.50)	47.50 (37.50–72.50)	0.96	0.93–0.98
sports and recreational activities	25.00 (12.50–50.00)	25.00 (6.25–43.75)	0.94	0.90–0.97
total	221.47 (133.01–287.57)	207.72 (136.69–270.88)	0.97	0.94–0.98

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
