# Peer review of "Translation, Cross-Cultural Adaptation, and Psychometric Properties of the Polish Version of the Hip Disability and Osteoarthritis Outcome Score (HOOS)"

_medicina, 2019, doi:10.3390/medicina55100614_

Round 1

Reviewer 1 Report

The many abbreviations in the abstract are not appropriate at all - please, change them with the full terms.

Sentence of 'The instruments used to 50 assess HRQoL are based on patients' opinions...' is the new paragraph explaining about assessment tools for the issue. More other available similar tools and explanation on the HOOS preference should be explained. 

Line 54 - where the instrument was to develop

Line 55 - what is WOMAC?

Line 56 - grammar mistakes (is - missing)

Line 57 - what languages - better to mention in the text. And what this means for you and for validation process? 

Line 75 - 'comprising a large group of older patients and healthy volunteers.' it is not clear was the age structure of both groups similar? if no - your explanation.

Line 75-76 - please, add ethical approval information in the separate paragraph as ethical considerations.

Line 83 - the inclusion criteria presented for two groups in one section is confusing - I would suggest to separate them.

this has to be re-written: 'having other barriers to participation, like 92 being alcohol or substance dependent within six months - How do you know that? I am sure this is difficult to determine precisely, or current alcohol or substance abuse; with an amputated limb; cardiac or other medical instability - what exactly? CD...others?; immobilized; fractured; having active malignancy; mental illness.

Translation procedure Line 96-100 - I am sure the much more information from the results part have to appear here. In the results part you should present more what the issues arised and how  final consensus was achieved, what was corrected and changed, ect.

Line 103-103 - information is much repeatable along the manuscript...you should reduce that.

Line 105-106 - style and syntax of sentence is not correct.

Line 120-121  -how you prove that? And how many patients participated in test-retest?

The the procedure section it would be logical to present the instruments first and then talk about the work done on them.

Subsection 2.2.2. is mixed with test information and statistical one, eg. the range of appropriate level of reliability. I would suggest to reconsider what information on tests you want to present here and all other test assessment information present in Statistical part. And not all ranges are presented, eg, how you assessed test-retest? Or what range and its interpretation of ICC was used?

Are all of HOOS items positively formulated - are any reverse items? 

141 - finally the full term of WOMAC appears!  -you should move this much ahead.

165-166 what is the reason to state the test once again if you already presented it before. Here the range and interpretation should appear.

In overall, the statistical information is too detailed and have to be condensed. 

174 - what guidelines exactly and checklists? The references are spokeless. 

177-179 this is only one example how the text should be improved and not-repetitive: you mention Polish twice here.

Translation process is very important but information here is very sparse on competency of translators, especially those of back translation.

189-190 - what exactly? it is very important to mention all the difficulties citing the items or some problematic term translation, ect.....

191-192 - do you speak about face validity of instrument here?

193-195 - what exactly do you mean? what items/terms/ect from SF was helpful and how?

197 - who were those experts? how many?

197-203 again - nothing specific about lessons learned on translation, terms, wording, cultural appropriateness in order to share information for future similar work

section 3.2. - it is really hard to read all this information  -better to use a table.

229-230 - repeatable information again

3.3.2 - in case the instrument has domains - multidimensional tool, it is much correct to provide the internal consistency measurements inside of each domain as well.

tables 1 and 2 and 4 have to be joint together for the best layout of the paper as they report on the same score.

3.3.5. and table 4 -  You made comparative analysis (was it paired sample?) but simple correlation (Spearman Brown) of both tests would be efficient.

The important question - why FA was not decided to confirm the structure or to determine the polish structure in order to detect cultural influence? Such decision has to be discussed anyway.

269-271 - the repeated information from the introduction.

278-284 - here information is confusing? when the Polish instrument was prepared? 10 years ago? When the study was conducted? also 10 years ago?

If CLEAR project is important here, you have to say more about that.

286 - the total alfa is usually high but it is more important to see the separate domains of the measure.

300-303 - here you have to be kmuch clear what you want to state. Further sentence in the next paragraph is almost the same what you state here.

In overall, the discussion has to be reviewed once again and information condensed not repeating the results so much. The reference list seems to be rather long for psychometric paper - please, reconsider all redundant sources.

The same for conclusions - you repeat the results and discussion here instead of necessary  interpretation 335-337.  The further lines are good.

Author Response

Response to Reviewer 1 Comments

Dear Reviewer,

Authors are thankful for the patient and careful review of the manuscript. The manuscript was corrected accordingly to the reviewer’s remarks.

The authors would like to assure that every effort was undertaken to address all issues raised by the reviewers honestly and reliably.

The responses to detailed remarks are presented below.

-Reviewer 1
The introduction, research design description, methods, results, and conclusions were improved using wise and appropriate reviewer’s remarks.

The many abbreviations in the abstract are not appropriate at all - please, change them with the full terms. 

Response: Abbreviations were changed into the full terms,

Sentence of 'The instruments used to 50 assess HRQoL are based on patients' opinions...' is the new paragraph explaining about assessment tools for the issue. More other available similar tools and explanation on the HOOS preference should be explained. 

Response: Corrected as suggested

Line 54 - where the instrument was to develop

Response: The information is added.

Line 55 - what is WOMAC?

Response: Full terms are added and the abbreviation. All used instruments are described in the subchapter dedicated to questionnaires. Transposing the single paragraph would make the content nonsystematic. We would also appreciate understanding the decision to keep the description of the WOMAC questionnaire as it is.

Line 56 - grammar mistakes (is - missing)

Response: corrected

Line 57 - what languages - better to mention in the text. And what this means for you and for validation process? 

Line 75 - 'comprising a large group of older patients and healthy volunteers.' it is not clear was the age structure of both groups similar? if no - your explanation.

Line 75-76 - please, add ethical approval information in the separate paragraph as ethical considerations.

 Response: corrected

Line 83 - the inclusion criteria presented for two groups in one section is confusing - I would suggest to separate them.

 Response: Inclusion criteria are separated as suggested

this has to be re-written: 'having other barriers to participation, like 92 being alcohol or substance dependent within six months - How do you know that? I am sure this is difficult to determine precisely, or current alcohol or substance abuse; with an amputated limb; cardiac or other medical instability - what exactly? CD...others?; immobilized; fractured; having active malignancy; mental illness.

 Response: Recruited patients/subjects were interviewed to confirm their abilities for participation in the study.

Translation procedure Line 96-100 - I am sure the much more information from the results part have to appear here. In the results part you should present more what the issues arised and how  final consensus was achieved, what was corrected and changed, ect.

Response: The purpose of this subparagraph was informative how far the COSMIN guidelines were followed. The extensive protocol explanation is available in the literature, so we added the citation.

Line 103-103 - information is much repeatable along the manuscript...you should reduce that.

Response: The information is reduced, as suggested.

Line 105-106 - style and syntax of sentence is not correct.

Response: The sentence is corrected.

Line 120-121  -how you prove that? And how many patients participated in test-retest?

Response: The sentence is corrected. The number of patients who participated in the test-retest is given here and in the results chapter.

The the procedure section it would be logical to present the instruments first and then talk about the work done on them.

Response: The procedure section is improved, as suggested.

Subsection 2.2.2. is mixed with test information and statistical one, eg. the range of appropriate level of reliability. I would suggest to reconsider what information on tests you want to present here and all other test assessment information present in Statistical part. And not all ranges are presented, eg, how you assessed test-retest? Or what range and its interpretation of ICC was used?

Response: We do appreciate this suggestion. The statistical information is moved.

Are all of HOOS items positively formulated - are any reverse items? 

Response: Items were formulated by the developers in 2003, as cited in the text with no reverse items. The aim of our research focused on the translation of the questionnaire and its validation.

141 - finally the full term of WOMAC appears!  -you should move this much ahead.

Response: The full term is moved ahead.

165-166 what is the reason to state the test once again if you already presented it before. Here the range and interpretation should appear.

Response: The fragment was moved to the statistical part, as suggested.

In overall, the statistical information is too detailed and have to be condensed. 

Response: The statistical information was compacted and presented in details in the statistical subchapter as suggested. The statistical information includes the used statistical methods only. The detailed information about it is necessary to better understand of obtained results and conclusions.”

174 - what guidelines exactly and checklists? The references are spokeless. 

Response: Utilized guidelines are mentioned and cited.

177-179 this is only one example how the text should be improved and not-repetitive: you mention Polish twice here.

Response: The sentence is corrected.

Translation process is very important but information here is very sparse on competency of translators, especially those of back translation.

Response: The back translation process details are added.

189-190 - what exactly? it is very important to mention all the difficulties citing the items or some problematic term translation, ect.....

Response: The sentence is corrected.

191-192 - do you speak about face validity of instrument here?

Response: The sentence is changed and cleared.

193-195 - what exactly do you mean? what items/terms/ect from SF was helpful and how?

Response: The meaning is corrected – the SF-36 translation into Polish served as an example to be followed.

197 - who were those experts? how many?

Response: The committee for this translation was described above.

197-203 again - nothing specific about lessons learned on translation, terms, wording, cultural appropriateness in order to share information for future similar work

section 3.2. - it is really hard to read all this information  -better to use a table.

Response: The descriptive statistics is left as before due to the typical presentation of such data in most of the articles concerning similar topics. 

229-230 - repeatable information again

Response: Redundancy is removed

3.3.2 - in case the instrument has domains - multidimensional tool, it is much correct to provide the internal consistency measurements inside of each domain as well.

Response: The information about separate domains is available in the table 2 and in the text “Moreover, all domains showed a significant correlation with the total score of the HOOS, ranging from r = 0.84, p-value < 0.05 (for HOOS quality of life) to 0.95, p-value < 0.05 (for HOOS daily living)”

tables 1 and 2 and 4 have to be joint together for the best layout of the paper as they report on the same score.

Table 4. The results of the test-retest reliability of the HOOS Polish version.

3.3.5. and table 4 -  You made comparative analysis (was it paired sample?) but simple correlation (Spearman Brown) of both tests would be efficient

Response: Yes, it was a paired sample. The Spearman-Brown statistic is used to estimate the reliability of the entire test based on the reliability of its half. It is used in the split-half method - one group of individuals split in half. We used a test-retest method, i.e., the same group of individuals was tested twice. So, a better solution seems to be a comparative analysis.

The important question - why FA was not decided to confirm the structure or to determine the polish structure in order to detect cultural influence? Such decision has to be discussed anyway.

Response: Factor analysis (FA) was not used in this study. FA is a statistical method used to describe variability among observed, correlated variables in terms of a potentially lower number of unobserved variables called factors. In psychometrics, it was usually effective in general mental ability, or the field of intelligence research. Authors did not anticipate intelligence or mental abilities problems due to clear enrollment criteria to the study.

269-271 - the repeated information from the introduction.

278-284 - here information is confusing? when the Polish instrument was prepared? 10 years ago? When the study was conducted? also 10 years ago?

Response:

If CLEAR project is important here, you have to say more about that.

Response: The full term and citation are added.

The CLEAR project was the spark to start this research to use an accurate hip instrument. CLEAR  (Clinical  Leading Environment for the Assessment of Rehabilitation protocols in home care) project was funded under the  ICT  Policy  Support Programme – CIP-ICT-PSP-2007.2.2 –ICT for Ageing Well. (https://www.signomotus.it/brochure/pdf_en/CLEAR_Final_Results_of_the_Project.pdf)

A large scale Pilot study has been conducted in four Member States of the  European  Union  (Italy,  Spain,  The  Netherlands,  Poland)  to demonstrate the feasibility of a Tele-rehabilitation service using Habilis, a general-purpose software platform. Patients received customized telerehabilitation sessions and executed the specific session exercises at home or in the nearest kiosk. Exercises according to medical team’ (physician/surgeon and physiotherapists) prescriptions performed by patients were recorded and assessed remotely and asynchronously. The  Center of Excellence “TeleOrto” hip and knee telerehabilitation protocols were implemented for patients for at-home exercises before and after Total Joint Replacement. The HOOS and other outcomes instruments were selected to use to assess the outcomes.

286 - the total alfa is usually high, but it is more important to see the separate domains of the measure.

Response:

300-303 - here you have to be kmuch clear what you want to state. Further sentence in the next paragraph is almost the same what you state here.

Response: The young adults' group was as the general comparison. The approach can be compared to the very typical statistical approach used in densitometry or other laboratory findings where young adults values are used to set the normative values. 

In overall, the discussion has to be reviewed once again and information condensed not repeating the results so much. The reference list seems to be rather long for psychometric paper - please, reconsider all redundant sources.

Response: We believe that the discussion is improved. The reference list was analyzed and corrected.

The same for conclusions - you repeat the results and discussion here instead of necessary  interpretation 335-337.  The further lines are good.

Response: corrected as suggested.

Reviewer 2 Report

This is a study evaluating the validity and reliability of the Polish version of HOOS.

The paper is well written, however, I still have some major points:

Is it possible to provide some background knowledge about different definitions of validity (content, contract etc.) in the section Methods? Many studies quantify construct validity by using factor analysis. Since factor analysis gives indication about whether the classification of subscale structure is valid (meet the primary research idea). In contrast, correlational validity was often used to quantify if two different scores measure the same things. Poor correlation does not mean a poor validity. It reflects only that the two scores measure different contents. Therefore, correlational validity examines actually the content validity rather than construct validity. However, the author define it to be a construct validity. Why?

HOOS total score correlated positively with WOMAC subscales, but negatively with WOMAC total score. Why? Some explanation should be given to this point.

Please provide the proportions of obesity for cases and controls. Please use BMI instead of weight and height in the analysis.

Some intensive discussion should be given about why there is a high correlation between HOOS and WOMAC, but lower correlation between HOOS and SF-36. I think, the most important point is that HOOS and WOMAC were specially developed for outcome measurements of hip OA. However, SF-36 was developed for assessing QoL for all kinds of patients, and not only for hip OA. Since the three scores have different focus and target population, their contents cannot be expect to be comparable with each other in all aspects. The higher correlation with WOMAC, but lower correlation with SF-36 indicate exactly that HOOS is a very specific instrument for hip OA. Lower correlation here does not mean lower validity.

The authors discussed that one of the strength of this study is the “homogeneous group of patients with OA of the hip joint”. I suppose, this is a limitation rather than strength. Since the use of homogeneous group of elderly, obese, severe cases of hip OA in this study will limit to present the true discriminative power of this instrument. This point should be mentioned in the discussion.

Author Response

Response to reviewer 2:

Dear Reviewer,

Authors are thankful for the patient and careful review of the manuscript. The manuscript was corrected accordingly to the reviewer’s remarks.

The authors would like to assure that every effort was undertaken to address all issues raised by the reviewers honestly and reliably.

The responses to detailed remarks are presented below.

Response: English language and style were checked.

Is it possible to provide some background knowledge about different definitions of validity (content, contract etc.) in the section Methods?

Response: The construct validity can be understood as "the degree to which a questionnaire measures what it claims, or purports, to be measuring."  Construct validity is demonstrated by using correlations to verify the relevance of the questionnaire's elements. Questions from similar questionnaires, that has been found reliable were correlated with questions from the HOOS questionnaire under examination to determine if construct validity is present. Highly correlated scores are called convergent validity.  The presence of the convergent validity means that the construct validity is supported.

Many studies quantify construct validity by using factor analysis. Since factor analysis gives indication about whether the classification of subscale structure is valid (meet the primary research idea). In contrast, correlational validity was often used to quantify if two different scores measure the same things. Poor correlation does not mean a poor validity. It reflects only that the two scores measure different contents. Therefore, correlational validity examines actually the content validity rather than construct validity. However, the author define it to be a construct validity. Why?

Moreover, there is no single best way to evaluate construct validity. It could be demonstrated by using a lot of methods, such as content analysis, correlation coefficients, factor analysis, ANOVA studies demonstrating differences between different groups or test/re-test intervention studies, multi-trait/multi-method studies, etc. There is also no single best way to study the content validity. Procedures used to evaluate content validity can be classified generally as judgmental or statistical. Statistical methods include procedures such as multidimensional scaling, cluster analysis, or factor analysis.

Finally, the subchapter concerning the methodology has been rewritten.

HOOS total score correlated positively with WOMAC subscales, but negatively with WOMAC total score. Why? Some explanation should be given to this point.

Response: The explanation was inserted in table 3 - the total score outcome measure is transformed in a worst to best scale (100 - no symptoms and 0 - extreme symptoms). The transformation of the scale was applied according to the HOOS Scoring protocol (www.koos.nl).

Please provide the proportions of obesity for cases and controls. Please use BMI instead of weight and height in the analysis.

Response: We used BMI instead of the weight and height analysis, as suggested.

Some intensive discussion should be given about why there is a high correlation between HOOS and WOMAC, but lower correlation between HOOS and SF-36. I think, the most important point is that HOOS and WOMAC were specially developed for outcome measurements of hip OA. However, SF-36 was developed for assessing QoL for all kinds of patients, and not only for hip OA. Since the three scores have different focus and target population, their contents cannot be expect to be comparable with each other in all aspects. The higher correlation with WOMAC, but lower correlation with SF-36 indicate exactly that HOOS is a very specific instrument for hip OA. Lower correlation here does not mean lower validity.

Response: This point is mentioned in the discussion, as suggested.

The authors discussed that one of the strength of this study is the “homogeneous group of patients with OA of the hip joint”. I suppose, this is a limitation rather than strength. Since the use of homogeneous group of elderly, obese, severe cases of hip OA in this study will limit to present the true discriminative power of this instrument. This point should be mentioned in the discussion.

Response: The suggested limitation was added to the “Limitations” subchapter.

Round 2

Reviewer 1 Report

I am happy you find comments helpful to improve your manuscript. 

Author Response

Thank you very much for the generous comment.

Reviewer 2 Report

there is maybe a small mistake:

Line 336: "However, the lower correlation may necessarily mean lower validity."

it should be: "However, the lower correlation may not necessarily mean lower validity".

Author Response

Thank you for the careful and constructive review.